# Investigation of Resistance Genes in Genus *Vigna* Reveals Highly Variable NLRome in Parallel Domesticated Member Species

**DOI:** 10.3390/genes14061129

**Published:** 2023-05-23

**Authors:** Jehanzaib Asif, Fatima Qureshi, Muhammad Zain, Hamza Nawaz, Effat Naz, Shahid Fareed, Aqsa Bibi, Sehar Nawaz, Fozia Saleem, Muhammad Shafique, Saba Tabasum, Umer Maqsood, Saad Serfraz, Saad Alkahtani

**Affiliations:** 1Evolutionary Biology Lab, Centre of Agricultural Biochemistry and Biotechnology, University of Agriculture, Faisalabad 38000, Pakistan; jahanzaib.aj.066@gmail.com (J.A.); fatimaqur99@gmail.com (F.Q.); muhammad.zain1197@gmail.com (M.Z.); hamzanawaz100100100@gmail.com (H.N.); 2017bsbote0041@student.gudgk.edu.pk (S.F.); seharnawazpbg@yahoo.com (S.N.); 2Metabolomics Innovative Institute, University of Alberta, Edmonton, AB T6G 2R3, Canada; fozia@ualberta.ca; 3Sheikh Zayad Hospital, Rahim Yar Khan 64200, Pakistan; 4Department of Plant Breeding and Genetics, College of Agriculture, University of Sargodha, Sargodha 40100, Pakistan; saba.tabasum@uos.edu.pk; 5Agricultural Biotechnology Division, National Institute for Biotechnology and Genetic Engineering, Constituent College of Pakistan, Institute of Engineering and Applied Sciences, Faisalabad 38000, Pakistan; umermehar10@gmail.com; 6Department of Zoology, College of Science, King Saud University, P.O. Box 2455, Riyadh 11451, Saudi Arabia; salkahtani@ksu.edu.sa

**Keywords:** TIR genes, *Vigna*, NLR genes, diploid evolution, novel resistance genes

## Abstract

*Vigna* is a unique genus that consist of multiple crop species that are domesticated in parallel fashion between 7–10 thousand years ago. Here we studied the evolution of nucleotide-binding site leucine-rich repeat receptor (NLR) genes across five crop species of genus *Vigna*. In total identified 286, 350, 234, 250, 108 and 161 NLR genes were from *Phaseolous vulgaris*, *Vigna. unguiculata*, *Vigna mungo*, *Vigna radiata*, *Vigna angularis* and *Vigna umbellata* respectively. Comprehensive phylogenetic and clusterization analysis reveals the presence of seven subgroups of Coiled coil like NLRs (CC-NLR) genes and four distinct lineages of Toll interleukin receptor like NLRs (TIR-NLR). Subgroup CC_G10_-NLR shows large scale diversification among *Vigna* species suggesting genus specific distinct duplication pattern in *Vigna* species. Mainly birth of new NLR gene families and higher rate of terminal duplication is the major determinants for expansion of NLRome in genus *Vigna*. Recent expansion of NLRome in *V. anguiculata* and *V. radiata* was also observed which might suggest that domestication have supported their duplication of lineage specific NLR genes. In short, large scale difference in the architecture of NLRome were observed in diploid plant species. Our findings allowed us to hypothesized that independent parallel domestication is the major drivers of highly divergent evolution of NLRome in genus *Vigna*.

## 1. Introduction

Genus *Vigna* is an important member of the family *Fabaceae* and consist of multiple crop species which are under cultivation in diverse geographical regions of Africa, Middle East Asia, South Asia, East and South east Asia [1]. It’s a pantropical genus containing 104 species widely distributed in tropical and subtropical regions [1,2]. Five major crop species including *Vigna unguiculata*, *Vigna angularis*, *Vigna mungo*, *Vigna radiata* and *Vigna umbellata*. Interestingly all five species have undergone parallel yet independent domestication events. In case of *V. unguiculata*, also known as Cowpea, was domesticated in Africa and latter spread across Asia, South, central and North America and Europe [3]. *V. unguiculata* is known for Its resilience for abiotic and biotic stresses. *V. angularis* also known as adzuki bean, was domesticated in China ~12,000 years ago and its also known for broad adaptability and high tolerance to poor soil fertility [4]. *V. mungo* (black gram) and *V. radiata* are domesticated in India around 7000 years ago and cultivated mostly in South, East, southeast Asia [5,6]. *V. umbellate* is also known as rice bean, it’s a lesser-known pulse and also domesticated in India [7]. *Vigna* crops are ideal candidates to study the effect of parallel domestication. It`s been reported that these cultivated species does harbor major and minor domestication-related QTLs [8,9]. However, domestication often leads to narrowing of genetic diversity resulting in compromised adaptability towards abiotic and biotic stresses.

Plant response to various biotic stresses using disease resistance gene as a part of their effector triggered immune (ETI) response [10]. Among them Nucleotide binding site leucine rich repeat receptors (NLRs) are responsible for recognizing a pathogen’s effector through direct or indirect interactions, which triggers a number of defensive mechanisms, including hypersensitive response or localized programmed cell death [11]. There are four major subclasses are divided on the basis of distinct N-terminal domain fusion (1) TIR-NLR subclass containing Toll/interleukin-1 receptor (TIR domain) (2) CC-NLR is the most diversified class containing coiled coil in the N-terminal (3) CC_R_-NLR are considered as helper NLRs (4) CC_G10_-NLR containing distinct type of CC and forming a monophyletic group. The wide range of pathogenic effectors are recognized because of large scale of sequence variability of NLR genes. Several mechanism are involved in the evolution of NLR genes including diversification of binding specificities in LRR domain, which is responsible for direct protein-protein interaction with the pathogens effectors [12,13]. Recent studies suggest that NB-ARC domain regions are evolving under negative selection, however LRRs are evolving under positive selections. Gene conversion, tandem duplication followed by local rearrangements are the major mechanism that contribute to the recognition of distinct pathogenic effectors [12,13]. As the NLR gene family continuously evolves, the regulation of their expression must also evolve to adapt to the changing environment and new virulent strains of pathogens.

The NLR gene family is highly variable in plant genomes and evolved through duplication, crossing-over, recombination, and gene conversion events [14,15]. Positive diversifying selection, which favors mutations, is often seen in NLRs, contributing to the gene family’s high variability [14]. Variation in NLR genes family occurs at distinct levels, such as allelic variation, presence/absence variation and copy number variations (CNVs). NLR copy numbers can vary significantly among species, and these genes tend to be overrepresented in regions with presence/absence variations [12]. This evolutionary response can be related to the severity and frequency of pathogen infection and the variability of NLR presence with the occurrence of effectors in pathogen populations [16]. Variations have also been associated with a lower accumulation of polymorphism, especially for NLRs involved in indirect recognition where subtle changes to the NLR can trigger autoimmunity. To this date, no study is reported to understand the evolution of NLR genes in *Vigna* species [17,18].

Here in this study, we have utilized the NLRtracker for the mining and annotation of NLR genes. Subsequently, we have utilized comprehensive evolutionary approaches to under the distinct evolution of NLR genes in genus *Vigna*. Since members of genus lack polyploidy except one species, all other species are diploid in nature [1,2]. Here we attempted to address the important question (1) Effect of parallel domestication on the evolution of NLRome. What are the main species that have shown expansion and contraction in the NLRome? What are the major mechanisms responsible for the expansion and contraction of NLR genes in *Vigna* species? whether domestication of *Vigna* species have narrowed the NLRome?

## 2. Materials and Methods

### 2.1. Mining of NLR Genes in Arachis Species

The genome sequence of multiple species of genus *Vigna* were available at different genome database portals. We selected NCBI genome portal to acquire the genome and annotation files for five *Vigna* spp. in order to retain the homogeneity in the sequence headers. We downloaded genome of *V. mungo*, *V. radiata*, *V. umbellata*, *V. angularis* and *V. unguiculata* (Appendix A). All genomes files were already annotated into respective transcriptome, proteome and gene transfer file format. The reference proteomes of all five *Vigna* genomes were subjected to NLRtracker pipeline [17,19]. The NLRtracker output files provides domain configuration, complete sequence all NLR gene along with conserved NB-ARC domain sequences from each identified sequence. Since several entries of NLR genes remained indeterminate using NLRtracker pipeline therefore manual curation using clusterization and phylogeny.

### 2.2. Phylogeny and Classification of NLR Genes

All known characterized NLR genes were acquired from PRG database libraries of NB-ARC domains [20]. These NB-ARC domain were clustered at 50% of sequence identity using UCLUST tool [21,22]. The nomenclature provided by Eunyoung Seo et al. [10] were adapted for NLR genes identified in family *Fabaceae* using clusterization and phylogentic approaches using seed probes. NB-ARC domains from *Vigna* species were aligned with seed probes from reference NLR genes using using MUSCLE (version 1.26) to build comprehensive phylogeny. Conducting the maximum likelihood analysis through IQtree v 2.0 [23], the most appropriate model of evolution (−m JTT + F + R10) was chosen, with 1000 bootstrap replicates being incorporated in the process.

### 2.3. Gene Density of NLR Genes across Vigna Chromosomes

Gene density maps were built using three file GFF3, NLR output file from NLR tracker and assemble genome file. Where assembled genome file was used to build fasta index. We then utilized to intersect NLR genes on GFF3 file using BEDtools [24] with bin size of 5 Kb. Resulting count files were then manually edited by providing bin number to each coordinate and built of karyotype file for R package RIdeogram [25].

### 2.4. Evolutionary Analysis in Vigna NLRs

Custom build pipeline using bash script was developed to perform multiple steps of evolutionary analysis including utilization of Clustalw to align protein sequence from each subgroups (Li, 2003). Subsequently, pal2nal software (version 1.0) was linked in the pipeline to perform protein guided nucleotide alignment [26]. At the end gaps were removed and substitution rate ks were estimated using MA method of tool ka/ks calculator and selection value on each paralog pair was tested using Fisher test and significant parolog pairs were kept and duplication events with *p*-values lower than 0.01 were not included [27]. In order to study orthologous relationship of NLR genes in *Vigna* species Ortho-venn2 was utilized [28]. Default settings were utilized except E-value of 10 × 10^−2^. For understanding NLR gene birth and death, Orthofinder was utilized using all identified NLR genes from Vigna species [29]. Ortholog sequences between the *V. unguiculata*, *V. mungo*, *V. umbellate*, *V. radiata* genomes were also obtained from Orthofinder [29]. Output species tree was converted into ultrametric tree using R package APE and the orthogroups were manually labelled [30]. Both the orthogroups and ultrametric tree files were used as input for CAFE5 [31], and gene gain and loss at each node was performed by parsing the resulting files.

### 2.5. Expression Analysis of NLR Genes Using RNA-Seq

We further determined the basal expression of identified NLR genes in available transcriptome datasets of *V. radiata* and *V. unguiculata*. Whole transcriptome based assay are sparingly available due to paucity of scientific attention provided for legume species. Here we screened two transcriptome dataset with significant alignment score. For *V. radiata* dataset was acquired under bioproject number PRJNA779876 that includes 12 samples that compared the expression between two tissues shoot apex and young pod. Secondly, we further screened basal expression levels of NLR genes in *V. unguiculata*, where in total 28 sample were used to compared the expression between different tissues. We utilized the HISAT, Stringtie and Ballgown based alignment and quantification pipeline as already mentioned in previous article [22,32].

## 3. Results

### 3.1. Gene Mining of NLR Genes in Vigna Species

We have utilized two approaches for the identification of resistance genes in the five species of this genus *Vigna*. Primarily we utilized the NLRtracker pipeline for NLR genes mining and successive annotations. Closest relative *Phaseolous vulgaris* was considered as the closest outgroup and selected for comparison of NLRome diversity. In total we identified 286, 350, 234, 250, 108 and 161 NLR genes were identified from *P. vulgaris*, *V. unguiculata*, *V. mungo*, *V. radiata*, *V. angularis* and *V. umbellata* respectively (Figure 1). Secondly, we further utilized manual clusterization of identified NLR genes to verify the number of NLR gene reported from each species. Substantial NLR genes and sub-class variations were observed among these five species of *Vigna*. Previous, investigations of NLR genes in genus *Cicer*, *Trifolium* and members of Dalbergioids [22] did not reported such variation within the diploid species as observed in case of genus *Vigna*. Large scale expansion of NLRome were observed for *V. unguiculata* and *V. radiata* where class TIR-NLR subclass have highest increase. Conversely, *V. angularis* have shown the significant contraction in the NLRome. Interestingly, number of CC_R_-NLR remained unchanged irrespective to expansion and contraction and their genome size. It reiterates the fact that helper NLR (CC_R_-NLR) are conserved among *Fabaceae* family. The repertoire of NLR genes identified here are appended in the Appendix A.

### 3.2. Gene Density among of NLR Genes in Vigna Species

Genus *Vigna* have shown substantial diversity in the repertoire of resistance genes in each species. We further studied the architecture of these genes in the assembled genome from each member species. Gene density maps of identified NLR genes were built using the bin size of 5 kb (Figure 2). These maps have shown that *V. umbellata* and *V. mungo* possess equal distribution of NLR genes across the genome, whereas the *V. radiata* and *V. unguiculata* possess large scale clusters of NLR genes. These clusters are believed to be develop as a result of tandem duplication of NLR genes due unequal crossing over, retro-transposition, gene conversion, asymmetric recombination [33,34]. In short, tandem duplication have played important role in expansion of NLRome in *V. radiata* and *V. unguiculata*. Due to lack of highly resolved chromosomally anchored genomes our estimated loci maps might show slight deviation to the actual loci map.

### 3.3. Phylogenetic Analysis and Classification of NLR Genes

Initially, reference NLR genes and identified NLRome from each species were clustered at 75 percent identity using CD-HIT and utilized for the manual classification of NLR genes. Representative members of NLR genes from each cluster from *V. unguiculata*, *V. mungo*, *V. radiata*, *V. angularis* and *V. umbellata* were utilized for the reconstruction of phylogenetic relationship (Figure 3). All five species have shown highly polyphyletic TIR-NLR and CNL-NLR subclasses. Especially five major distinct radiation of TIR-NLR were observed, where majority of putatively novel nodes belonged to *V. angularis* and *V. radiata*. In case of CNL-NLR six major sub-groups were observed G4, G7, G11, CNL-UN, CC_G10_-NLR and CC_R_-NLR. It is consistent with the previous observation that subgroups G1, G2, G3, G5, G6, G8 are not present in *Fabaceae* family However, significant divergence and expansion in G4 and G7 were observed in genus *Vigna*. That might suggest another hypothesis that other CC-CNL subgroups might be removed due to extinction of pathogens of Fabaceae member. As already indicated, helper NLRs (CC_R_-NLR) were found stable across all genomes and interestingly two major radiation were observed which are common to all five species. However, third radiation were discovered that was only present in *V. radiata* and absent in other members of this genus. Class CC_G10_-NLR have shown the highest phylogenomic diversity among all members of this species especially *V. radiata* and *V. angularis*. Three major lineages of this subclass were frequently observed for all five species of this genus. In short, NLRome of genus *Vigna* have shown highly unbalanced gene duplication occurrence that suggest that majority of duplication have occurred after speciation from common ancestor.

We also compared the selection pressure within in the pairs of paralogs from four major subgroups (G4, G7, CCG10-NLR and TIR-NLR). Highest selection pressure were observed for CC_G10_-NLR where selection pressure of 0.5 (median) and above were observed in all species especially *V. unguiculata* and *V. radiata*. However, the G4 and G7 have shown variable range of selection pressure between 0.4 to 0.5 (median). It suggest that tendency of divergence of CC_G10_-NLR is higher as compared to other subgroup diversification incase genus *Vigna* (Appendix A).

### 3.4. NLR Gene Birth and Death Ratio

Gene gain and loss is a prevalent phenomenon for NLR gene evolution, due to consistent arms race between host and pathogen. NLR gene identified from NLRtracker were further subjected to orthology based analysis (Figure 4). In total 46 core orthologs clusters were found as an ancestral NLRome of genus *Vigna*. Surprisingly, 20 unqiue ortholog clusters were observed in *V. unguiculata*. We further tested this unique finding by understanding gene gain and loss analysis using Orthofinder and CAFÉ analysis (Figure 4 birth and death of gene families (green/red) and number of genes duplicated colored in blue). In case of *V. unguiculata*, total 36 gene families were gained and recorded highest rate of terminal duplication, where 238 gene were gain as a result of gene duplication after speciation and reported a loss of 16 gene families. Similarly, *V. radiata* also have shown the gain of 41 distinct gene families and significant duplication of already present gene families. On the otherside, *V. angularis* and *V. umbellata* have shown extensive loss of distinct NLR families and slower rate of terminal duplication. Overall, since the divergence from the common ancestor (approx. 9–10 Mya), contraction of NLRome was observed and latter after complete speciation several species have shown expansion. In short, large scale expansion of *V. unguiculata* and *V. radiata* were due to large scale gain in gene families and higher terminal duplication.

### 3.5. NLR Gene Duplication Assay

Previous analysis suggest that terminal gene duplication is one of the major reason for the expansion of NLRome in *Vigna* species. Here we explored the duplication history of genus *Vigna* by comparing the Ks values between paralogs of each subgroups (Figure 5). Genus *Vigna* shows one of the highest rate mutation (6.5 × 10^−9^) among Fabaceae family [35]. The divergence from common ancestor of *V. unguiculata*, *V. mungo*, *V. radiata*, *V. angularis* and *V. umbellata* occurred 9–10 Mya ([36] time tree reference). Collective Ks values obtained from all five species have shown one major curve of duplication between 6–10 Mya (Ks: 0.08–0.12), suggesting common expansion of NLRome in *Vigna* genus occurred after speciation. In addition one peak was observed for *V. unguiculata* and *V. radiata* between 1–3 Mya (Ks: 0.02–0.05) that suggest that their large scale expansion occurred quite recently. Subgroup G4–G7, G10 CNL and TIR-NLR have been amplified quite rapidly during recent 1–3 Mya leading toward birth of novel genes after speciation.

## 4. Expression Analysis

We compared the basal expression of NLR genes in *V. radiata* (PRJNA779876), where the 6 whole transcriptome samples, each from shoot apex and young pods were sequenced. Comparative transcriptome expression analysis have only shown the expression of four NLR genes that belongs to subgroup G4-CNL and CC_R_-NLR (Appendix A). Overall the expression level was same in shoot apex and young pod tissues. However two genes (Vradi10g02490.1 and Vradi0405s0010.1) have shown higher expression in both tissues. Secondly, we compared the expression levels of NLRs in six different tissues (seeds, roots, stem, leaf, flower and pod) in *V. unguiculata*. In total 15 genes have shown a significant quantitative expression in all these tissue, which suggest that these five genes are constitutively expressed in all six different tissues type of *V. unguiculata*. It should be noted that sharp increase in expression levels were observed suggesting that these genes are actively expressed. Genes that have shown active expression in both species belonged to subgroups G4-CNL and CC_R_-NLR (Figure 6).

## 5. Discussion

Disease resistance genes especially NLR genes are directly or indirectly interacting with pathogen effector molecules for coordinated defense response [33]. Host-pathogens are under continuous arms race to suppress primary and secondary responses. For this several novel NLR genes tend to evolve as result of duplication, recombination, unequal crossing over and mutation. Extinction of pathogen often lead to death of several NLR genes to maintain host fitness [33]. Among *Fabaceae* family, several members have been investigated including *Dalbergioids*, genus *Cicer* and genus *Trifolium* [22,32,37]. One common observation, was the importance of crop wild relatives (CWRs) that contain broader and expanded NLRome and conversely domesticated crop species suffers from narrowing of genetic diversity [22,32,37]. Secondly, presence of polyploidy have considerably broaden and expanded the NLRome of allotetraploid species of *Arachis* and *Trifolium* [22,32]. However, genus *Vigna* only contain one polyploid species among 104 member species, yet its members contain highly diversified repertoire of NLR genes.

Findings of this study, suggest that *Vigna* genus have shown dramatic variation in their NLRome. Especially *V. radiata* and *V. unguiculata* have shown rapid expansion of NLRome whereas *V. umbellata* and *V. mungo* have shown slower rate of diversification. We employed comprehensive phylogeny, duplication assay, gene gain and loss analysis which suggest that diversification on NLR occurred after the speciation from common ancestor of *Vigna* between 9–10 Mya [2,36]. And birth of novel genes and terminal duplication are the major reason of the expansion on NLRome *V. radiata* and *V. unguiculata*. Duplication analysis further confirms that these two species have undergone a sharp expansion of NLRome during last three million years. In short all five species have shown species specific divergent evolution of NLR genes.

Since, *Vigna* species have undergone independent domestication in three major geographical regions. *V. radiata* and *V. unguiculata* have been domesticated in South Asia and Africa respectively, which could be the reason for their divergent NLR evolution. In addition, recent duplications of NLR genes in these two species allowed us to hypothesized that domestication have favored the expansion of NLRome. However, additional evidence will be required to test this hypothesis. Genome sequence of related wild relatives of *V. radiata* and *V. unguiculata* will be required to confirm whether the domestication has broaden the repertoire of NLR genes.

## 6. Conclusions

Investigation of NLR genes in genus *Vigna* reveals highly divergent evolution of this superfamily in the cultivated crop members. Common wave of NLRome expansion was observed after speciation from common ancestor. In addition, preferential duplication of CC_G10_-NLR have been observed in all five members of this genus suggesting a genus specific diversification. Birth of diverse gene families and terminal duplication are putatively main reason for the expansion of NLRome in *Vigna* species. In our opinion, lack of concerted evolution and specialized evolution of NLR genes occurred among the member of this genus. Recent expansion of NLRome in *V. anguiculata* and *V. radiata* might suggest that domestication have support their duplication.

## Figures and Tables

**Figure 1 genes-14-01129-f001:**
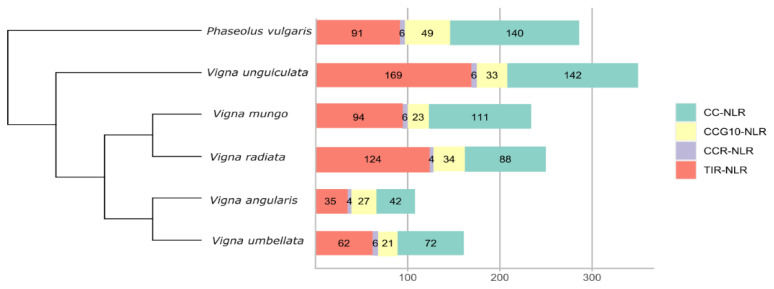
Distribution of NLR genes and its subclasses in *P. vulgaris*, *V. unguiculata*, *V. mungo*, *V. radiata*, *V. angularis* and *V. umbellata*.

**Figure 2 genes-14-01129-f002:**
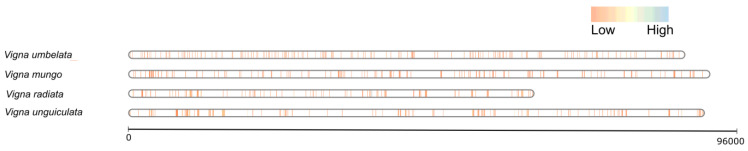
The NLR genes located on chromosomes are denoted in vertical blue and orange lines, inferring the gene density map between genomes assemblies of three *Vigna* species.

**Figure 3 genes-14-01129-f003:**
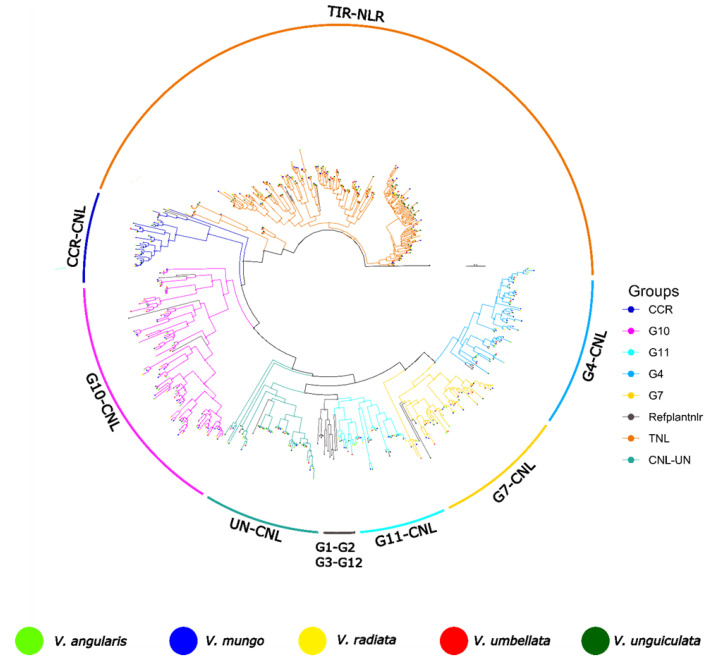
Reconstruction of phylogenetic relationship between NLR genes superfamily of genus *Vigna*, using *V. unguiculata*, *V. mungo*, *V. radiata*, *V. angularis* and *V. umbellate*.

**Figure 4 genes-14-01129-f004:**
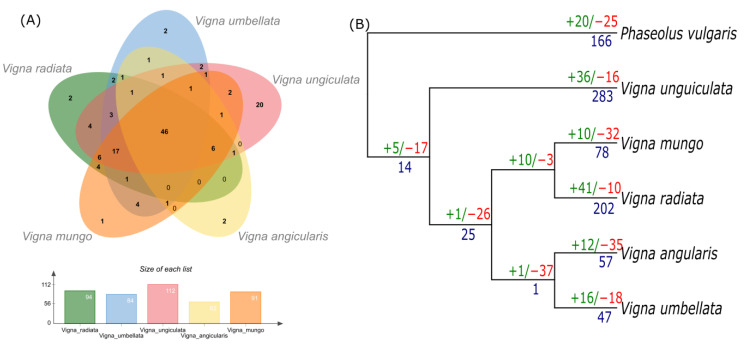
Orthology and birth/death analysis of NLR genes. (**A**) Venn diagram expressing common ortholog gene cluster found in five species of genus *Vigna*. (**B**) Phylogenetic tree expressing number of gained (green) and lost (red) gene families in member species. Number of duplicated genes are added in blue color.

**Figure 5 genes-14-01129-f005:**
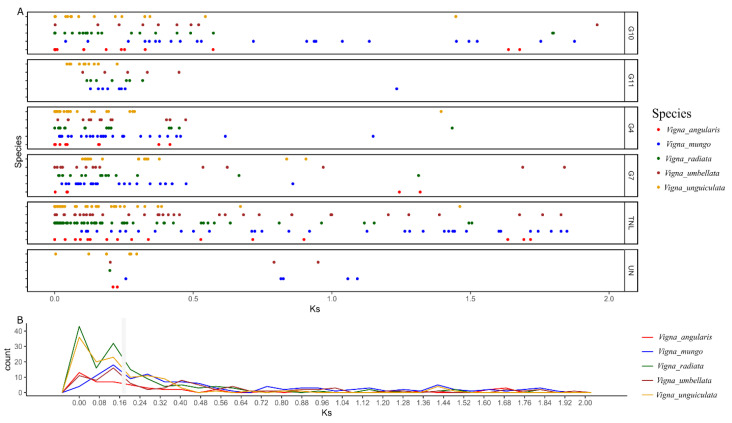
Estimation of NLRome duplication event in the genus Vigna. All five species belonging to the genus Vigna with Ks values of their paralogs are denoted. (**A**) The vertical axis indicate Ks value and horizontal axis shows frequencies. (**B**) The generalized observed duplication manner in all species.

**Figure 6 genes-14-01129-f006:**
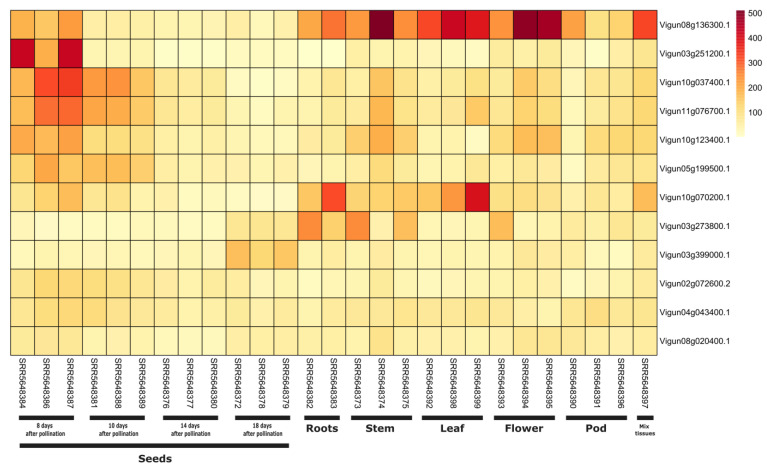
Comparative expression of NLR genes in 6 different tissues types of *V. unguiculata*. Here 13 identified NLRs have shown constitutive expression across all tissue types.

## Data Availability

All the accession numbers utilized in this study are provided. NLR genes identified in this study are included in the Appendix A.

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
