# Peer review of "Investigation of Resistance Genes in Genus Vigna Reveals Highly Variable NLRome in Parallel Domesticated Member Species"

_genes, 2023, doi:10.3390/genes14061129_

Round 1

Reviewer 1 Report

The authors analyzed several available genome bases and transcriptome datasets using diverse bioinformatics methods. They reported large scale difference in the totality of NLR genes in diploid Vigna species and their abundance and distribution of these genes. Using this evidence, the authors hypothesized that independent parallel domestication was the major driver of highly divergent evolution of these genes. While the scope of Vigna species was rather limited and direct proof for this hypothesis is missing, it seems quite plausible on our current knowledge about plant domestication and genome evolution. However, the authors should rather define their conclusions as putative and suggestive. The manuscript does not cite several important papers on divergence of NLR genes beyond Vigna, e.g., Solanum. While the study presents new data on chromosome localization of NLR genes and specific expression in plant tissues, it lacks any evidence on functional identification of NLR genes in Vigna species under study.

Minor editing of English language required

Author Response

Response to the reviewer 1

The authors analyzed several available genome bases and transcriptome datasets using diverse bioinformatics methods. They reported large scale difference in the totality of NLR genes in diploid Vigna species and their abundance and distribution of these genes. Using this evidence, the authors hypothesized that independent parallel domestication was the major driver of highly divergent evolution of these genes. While the scope of Vigna species was rather limited and direct proof for this hypothesis is missing, it seems quite plausible on our current knowledge about plant domestication and genome evolution. However, the authors should rather define their conclusions as putative and suggestive. The manuscript does not cite several important papers on divergence of NLR genes beyond Vigna, e.g., Solanum. While the study presents new data on chromosome localization of NLR genes and specific expression in plant tissues, it lacks any evidence on functional identification of NLR genes in Vigna species under study.

>>We thank reviewer for comments and suggestions made regarding this paper. Considering your concerns regarding conclusions, we have adapted our text to more suggestive nature rather than conclusive nature.

Secondly, member of family Fabaceae lacks attention of researchers. This paper is an attempt to understand the evolution NLR genes in genus Vigna. To this point we have screened 62 plant species of this family. We have not observed such diverse nature of NLR genes evolution in any other member of Family Fabaceae. I completely agree with you that this paper does not provide any functional characterization studies, but it does provide which are the most important subgroups of NLR genes that should be functionally characterized first from this Genus Vigna.

Reviewer 2 Report

1.     In line 17, delete the word identified.

2.     Reference no 27 is missing in the text.

3.     Arrange the references in ascending order in the text.

4.     Describe briefly the importance of NLR genes in the introduction.

5.     In line 74 abbreviate spp.

6.     In legend, for Figure 1: the scientific names should be in italics.

7.     In line 214, italicise the scientific names and genus.

8.     Is there any stress involved in the study? If yes mention them.

9.     Describe the evolutionary methodology in detail

10.  Kindly check the whole MS for typo errors and scientific names with minor English editing

Minor English language editing with scientific correctness is required

Author Response

All the changes with respect to reviewer 2 can be seen as tracked changes

Response to the comments made by Reviewer 2

We thank the reviewer for his positive comments. All the important changes are now addressed in the revised version of manuscript.

  1. In line 17, delete the word identified.

>> We appreciate raising important mistake, It now been addressed and be visualised under tracked changes.

  1. Reference no 27 is missing in the text.

>> missing reference is now added in the revised version,

  1. Arrange the references in ascending order in the text.

>> The references are now arranged in best possible manner.

  1. Describe briefly the importance of NLR genes in the introduction.

>> The additional and supplementary information regarding the importance of NLR genes is now added in the introduction from line 59 to line 68.

  1. In line 74 abbreviate spp.

>> Modified as suggested. Its now line number 99.

  1. In legend, for Figure 1: the scientific names should be in italics.

>>Modified as suggested.

  1. In line 214, italicise the scientific names and genus.

>> Modified as suggested.

  1. Is there any stress involved in the study? If yes mention them.

>> Thanks for pointing out an important question. The transcriptome study utilized are not from stressed samples. We only studied the basal expression of NLR genes in order evaluate the expression on identified NLR genes.

  1. Describe the evolutionary methodology in detail

>> Details are now mentioned in the evolutionary method section.

  1. Kindly check the whole MS for typo errors and scientific names with minor English editing

>> we have performed a detailed check on the whole MS and corrected typos in scientific names and performed substantial English editinsdfafs
